# Can VLMs Play Action Role-Playing Games? Take Black Myth Wukong as a Study Case

## Abstract

Recently, large language model (LLM)-based agents have made significant advances across various fields. One of the most popular research areas involves applying these agents to video games. Traditionally, these methods have relied on game APIs to access in-game environmental and action data. However, this approach is limited by the availability of APIs and does not reflect how humans play games. With the advent of vision language models (VLMs), agents now have enhanced visual understanding capabilities, enabling them to interact with games using only visual inputs. Despite these advances, current approaches still face challenges in action-oriented tasks, particularly in action role-playing games (ARPGs), where reinforcement learning methods are prevalent but suffer from poor generalization and require extensive training. To address these limitations, we select an ARPG, "*Black Myth: Wukong*", as a research platform to explore the capability boundaries of existing VLMs in scenarios requiring visual-only input and complex action output. We define 13 tasks within the game, with 76.9% focusing on combat, and incorporate several state-of-the-art VLMs into this benchmark. Additionally, we will release a human operation dataset containing recorded gameplay videos and operation logs, including mouse and keyboard actions. Moreover, we propose a novel **VARP** (**V**ision **A**ction **R**ole-**P**laying) agent framework, consisting of an action planning system and a human-guided trajectory system. Our framework demonstrates the ability to perform basic tasks and succeed in 90% of easy and medium-level combat scenarios. This research aims to provide new insights and directions for applying multimodal agents in complex action game environments. The code and datasets will be made available at `https://varp-agent.github.io/`.

## 1 Introduction

In recent years, LLM-based agents have achieved significant breakthroughs across various fields (Deng et al., 2023; Gur et al., 2024; He et al., 2024; Wang et al., 2024a; Zhang et al., 2023b), particularly with the integration of tools and memory modules(Zhou et al., 2024), as seen in AutoGPT and Reflection (Yang et al., 2023; Shinn et al., 2023). Among these, applying LLM-based agents in video games has become one of the most popular areas of research.(Qian et al., 2024; Park et al., 2023; Li et al., 2024; Wang et al., 2023b; 2024b) These methods input information from video games into LLMs, which then undergo complex reasoning and integration through agent frameworks, ultimately producing keyboard and mouse commands that can directly interact with the game to complete tasks. Previous works have mostly focused on accessing video game APIs to read in-game environmental and action information. For instance, the framework proposed by Wang et al.(Wang et al., 2023a) has been successfully applied in the game Minecraft. Agents can achieve autonomous mining, building, and attacking enemies in the game. However, this approach does not align with how humans play games, and most games do not offer open APIs, which limits the widespread application of this method. Recently, the emergence of vision language models (VLMs) like GPT-4o has further enhanced the visual understanding capabilities of these agents, showcasing broader potential in mobile apps and games. For example, the Cradle framework (Tan et al., 2024) has been implemented in Red Dead Redemption 2 (RDR2). It directly uses game screenshots from RDR2 as input, rather than using an API to read game memory information. However, Cradle relies heavily on text-based guiding information in the game screenshots to create new skills. For tasks or games

Table 1: Comparison of several existing agents. Among them, "API" refers to the model's use of video game APIs to access in-game environmental and action information, whereas "Screen" indicates that visual understanding is derived solely from game screenshots.

| Agents | Agent Type | Game | Game Type | Environment |
|---|---|---|---|---|
| Reflexion (Shinn et al., 2023) | LLM-based | ALFWorld | Text-based Adventure | API |
| ReAct (Yao et al., 2023) | LLM-based | ALFWorld | Text-based Adventure | API |
| Voyager (Wang et al., 2023a) | LLM-based | Minecraft | Sandbox | API |
| CreativeAgent (Zhang et al., 2023a) | VLM-based | Minecraft | Sandbox | API and Screen |
| Cradle (Tan et al., 2024) | VLM-based | RDR2 | AAA Action Adventure | Screen |
| DQN (analoganddigital, 2021) | RL-based | Sekiro | AAA Action Role-Playing | Screen |
| Other Project (Cat, 2024; fange, 2024) | RL-based | BMW | AAA Action Role-Playing | Screen |
| **VARP (Ours)** | VLM-based | BMW | AAA Action Role-Playing | Screen |

with weak textual guidance, such as some action role-playing games(ARPG), Cradle is unable to leverage the effective performance of VLMs. For ARPGs, many researchers employ reinforcement learning methods, where penalties and rewards are predefined for specific tasks. After extensive training periods and numerous iterations, the trained agents can complete the given specific tasks. However, RL-based agents can only accomplish tasks within the environment they were trained in and find it challenging to transfer to other tasks. ARPGs contain a large number of specialized tasks, which pose a significant challenge for RL-based agents with poor generalization capabilities. We conducted a comparison of some representative methods in Tab. 1.

Thus, most of the existing research focuses on relatively simplified settings. This simplification arises primarily from two significant challenges: 1) **Immediate visual input.** Since environmental data is not always accessible through game APIs, learning from visual input becomes a more straightforward strategy, especially in AAA games (characterized by A lot of time, A lot of resources, A lot of money), where understanding the immediate visual input is crucial. 2) **Action-oriented tasks.** Action games are immensely popular among players; however, in this domain, RL-based agents still dominate, which require extensive training time and have poor generalization ability. For VLM-based agents, the game interfaces of ARPGs provide very few textual hints, and most of the actions need to be learned through experience and self-innovation. As a result, previous agents have found it challenging to extract effective guidance information from visual inputs.

In this paper, we will select the "*Black Myth: Wukong*," abbreviated as BMW, an AAA ARPG, as our research platform for extensive experimentation. We are dedicated to establishing a VLM-based agent framework to thoroughly investigate the capability boundaries of existing models (e.g. GPT-4o, Gemini) in scenarios requiring visual-only input and complex action output. Among them, visual-only input refers to the model making decisions solely by understanding and analyzing the game screenshot, while complex action output necessitates the model to perform intricate and continuous actions, such as precise operations in combat scenarios.

To achieve this goal, we define 12 tasks in the game "*Black Myth: Wukong*," with 75% of these tasks being combat-related. Several state-of-the-art VLM models, including GPT-4o, will be incorporated into this benckmark to comprehensively explore their performance boundaries. Subsequently, to advance the development of VLM-based agents in AAA action games, we will open-source a human operation dataset, which includes records of mouse and keyboard commands as well as gameplay recordings. Lastly, we innovatively propose a VARP(Vision Action Role-Playing) agent framework, consisting of an action planning system and a human-guided trajectory system. Specifically, the action planning system is responsible for generating action combos that are suitable for combat scenarios, while the human-guided trajectory system learns from human data via retrieval. Through extensive evaluations, our proposed framework demonstrates the capability to accomplish basic tasks such as picking up items and opening treasure chests, while also succeeding in 90% of esay and medium battles. We hope this research will provide new insights and directions for the application of multi-modal agents in complex action game environments. The main contributions of this paper are summarized as follows:

- **Benchmark**. We define 12 tasks based on the game "*Black Myth: Wukong*," with 75% of these tasks focused on combat. Several state-of-the-art VLM models, including GPT-4o, will be incorporated into this benckmark to thoroughly explore their capability boundaries.

- **Dataset**. We release a dataset containing recorded gameplay videos along with relevant operation logs, including mouse movements, clicks, and keyboard actions, which includes 1000 records.

- **Framework**. We propose a VARP agent framework, which comprises an action planning system and a human-guided trajectory system. Through these systems, the agent can execute complex action combos and learn from human operation.

## 2 RELATED WORK

### 2.1 LLM AND VLM-DRIVEN AGENTS

In recent years, various intelligent agents driven by large language models (LLM) and multimodal language models (VLM) have gradually come to the forefront, demonstrating immense potential in multitasking and autonomous learning. For LLM, Reflexion (Shinn et al., 2023) enhances the decision-making ability of language agents through a framework of linguistic feedback and self-reflection, allowing agents to autonomously reflect in the face of feedback signals and maintain contextual memory during task execution and decision-making processes. ReAct (Yao et al., 2023), on the other hand, emphasizes real-time information retrieval and strategy adjustment. By combining reasoning with action and interacting with external knowledge sources (such as the Wikipedia API), it adds dynamic information retrieval capabilities, providing greater interpretability and controllability. Voyager (Wang et al., 2023a) can explore and learn skills in an unsupervised manner within the Minecraft environment, continuously exploring the world, acquiring diverse skills, and making new discoveries without human intervention through a combination of automated courses, skill libraries, and iterative prompting mechanisms.

For VLM, CreativeAgent (Zhang et al., 2023a) focuses on creative tasks, employing multimodal generation to achieve the construction of complex structures. Its combination of an imagination module and controller enables efficient planning and execution based on free-form language instructions and the generated task details. Cradle (Tan et al., 2024) takes video images displayed on a screen as input, extracting text and visual information to make decisions through a workflow of "reflecting on the past, summarizing the present, and planning for the future," outputting control signals for keyboard and mouse interaction, allowing AI agents to interact with software like humans without relying on any internal APIs.

## 3 METHODOLOGY

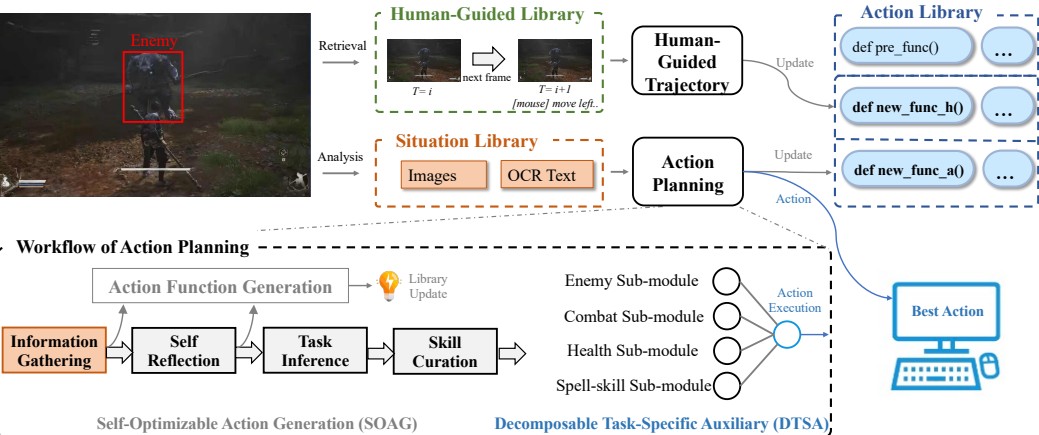

Figure 1: **Pipeline of VARP.** We propose a novel framework named the VARP agent, which directly takes game screenshots as input and generates keyboard and mouse operations to play the ARPG.

### 3.1 OVERVIEW

We propose a novel framework named the VARP agent, which directly takes game screenshots as input. Through inference by a group of Vision-Language Models (VLMs), it ultimately generates actions in the form of Python code, which can directly operate the game character. Each action is a sequence that consists of various combinations of atomic commands. These atomic commands include light attack, dodge, heavy attack, restore health, and others. Meanwhile, the VARP agent

maintains three libraries: a situation library, an action library, and a human-guided library. These libraries can be retrieved and updated to store intensive knowledge for self-learning and human guidance. Overall, the VARP agent is divided into two systems: the action planning system and the human-guided trajectory system, as shown in Fig. 1. In the action library, "def new_func_a()" represents the new action generated by the action planning system, while "def new_func_h()" represents the new action generated by the human-guided trajectory system. "def pre_func()" represents the predefined actions. The following sections will elaborate on each system in detail.

## 3.2 ACTION PLANNING SYSTEM

The action planning system is primarily used for action reasoning and generation. This system utilizes a situation library and an updatable action library as knowledge retrieval bases. Guided by input game screenshots, the system employs a group of VLMs to select or generate actions appropriate for the current situation. The generated situations and actions are stored or updated in the two libraries. Additionally, We propose decomposable task-specific auxiliary modules to break down large tasks into smaller subtasks, which are then distributed across multiple VLMs to reduce the occurrence of model forgetting and hallucinations. We also introduce a self-optimizable action generation module to encourage VLMs to generate new actions specific to some hard tasks, thereby completing complex tasks more efficiently and with higher quality.

### 3.2.1 BASIC VLMS GROUP

Inspired by Cradle (Tan et al., 2024), our main pipeline continues to adopt the five basic modules from Cradle, with some of these basic modules calling the VLM for reasoning, forming a basic group of VLMs. During initialization, we manually predefined some actions and placed them into the action library as the prior knowledge. Each action is a Python function with detailed textual annotations, and we computed the embeddings of these annotations for storage. **Information Gathering** is responsible for gathering information from sampled game screenshots, including textual and visual information related to situations and actions. The textual information primarily includes text guides, text labels, and notifications; the visual information mainly covers environmental positions, character actions, and interface icons. The former is assisted by OCR tools for text recognition, while the latter uses the object detection tools for visual localization. **Self Reflection** takes a few game screenshots from the last video in the situation library as input to assess whether the last executed action successfully produced the correct effect and whether the current task has been completed. If execution fails, the module needs to provide a reason for the failure to guide the next step in action generation. **Task Inference** infers the current task to be executed based on the results of previous modules, and generates the task description. **Skill Curation** calculates the similarity between the task description's embedding and the embeddings of the textual annotations in the action library to find some matching actions, which form the candidate action set. **Decision Making** utilizes the Chain of Thought (CoT) (Wei et al., 2023) approach to reason through and deeply analyze multiple sequential questions (such as whether to enable combat mode, restore health, or select from available spell skills, etc.). Finally, the module infers the most suitable action from the candidate action set, executes the Python code, and operates the keyboard and mouse to control the player character to complete the corresponding task. These five basic modules will record each intermediate product into the situation library.

### 3.2.2 SELF-OPTIMIZABLE ACTION GENERATION MODULE

The basic VLMs group can only acquire actions from a predefined action library or from game screenshots with clear textual prompts. For certain tasks in ARPGs that have weak textual guidance, such as real-time combat, this method is unable to learn new actions. Therefore, we propose a self-optimizable action generation module (SOAG) that allow the VARP agent to summarize the enemy's actions during combat, thereby optimizing existing actions and generating new ones to counter enemy attacks. The new actions are combinations of the two atomic commands: dodging and light strikes. The optimization goal is to maximize the evasion of enemy attacks and the ability to strike the enemy while minimizing the player character's health loss.

Specifically, in SOAG, we introduce a component responsible for action function generation. This component takes the information gathering and self reflection results, along with the current and

last game screenshots, as input. It analyzes the characteristics of the enemy under the current task, such as name, appearance, weapon, etc. Most importantly, it needs to analyze the enemy's current and previous actions. For example, for the hard enemy named Bullguard, its attack actions can be roughly categorized as: "charging forward with the axe", and "chopping the axe downwards three times consecutively", etc. Therefore, this component needs to inference new actions for dodging and counterattacking based on the current enemy actions. For instance, for "charging forward with the axe," the new action should be to dodge once and then attack continuously; for "chopping the axe downwards three times consecutively," the new action should be to dodge three more times before counterattacking. The generated new actions are permutations of the atomic operations "dodge" and "light attack." The generated actions are stored in the action library with detailed textual annotations.

### 3.2.3 DECOMPOSABLE TASK-SPECIFIC AUXILIARY MODULES

In ARPGs, especially in BMW game, the VLM's inference involves a large number of tokens, including multiple images and long texts. The attention mechanism used by VLMs allocates attention to all tokens in long texts. As the input length increases, the attention distribution becomes increasingly diluted. In the basic VLMs group, due to the excessive number of input tokens for each module, the model may fail to effectively focus on key information, leading to errors such as forgetting and hallucination. This issue is particularly evident in the decision-making module, where the VLM frequently makes mistakes when answering multiple questions.

To address this problem, we decomposed the basic modules and added multiple parallel auxiliary sub-modules for specific tasks, which are then integrated by the VLM. The structure is similar to an MLP. Specifically, as shown in the workflow of action planning in Fig. 1, we decomposed the original decision-making module that handled multiple tasks into 5 sub-modules. **1) Enemy Sub-module** is used to analyze the enemy's status (such as its health, position, etc.) and action description, which assists the agent in obtaining detailed information of the enemy. **2) Combat Sub-module** determines which combat method to use, including light attack or heavy attack, by observing the heavy-attack status in the bottom right corner of the game screen. **3) Health Sub-module** is responsible for constantly monitoring the player's health bar. If the health is consumed excessively, it assists the agent by prioritizing the action of recovering health. **4) Spell-skill Sub-module** monitors the status of the player's spell skills while simultaneously analyzing the situation in the combat state to determine the appropriate time to use available spell skills. **5) The integration sub-module** is responsible for integrating the outputs of all sub-modules and reasoning to determine the best action from the candidate action set for the current specific task. The decomposable task-specific auxiliary modules decompose long tokens and focus on each individual question, significantly improving the accuracy of the decision-making module.

### 3.3 HUMAN-GUIDED TRAJECTORY SYSTEM

Human actions are seen as valuable data, implicitly rich in knowledge of the physics and game world, which can lead to advanced action combinations for very complex tasks, such as way-finding tasks and high-difficulty combat tasks. To learn the human experience from this implicit data, we first collected a human dataset and then used it to improve the performance of our VARP agent. The collection process of human operation data and dataset analysis will be detailed in Sec. 3.1, which consists of mouse keyboard logs, and recording game screenshots. In this section, we focus on how to use it to implement a human-guided trajectory system. In this section, we refer to our annotated dataset as the human-guided library. It is a collection of pairs consisting of game screenshots and human operations, with each pair having a unique timestamp.

For very hard tasks in the game, we first take a screenshot of the current game interface. Based on this game screenshot, we query the human-guided library for the screenshot with the highest similarity. We then input this screenshot along with the subsequent n-frame screenshots and their corresponding operations into the human-guided trajectory system. This system will utilize a VLM to analyze and summarize the input images and operations, ultimately outputting a new human-guided action, which is then stored in the action library for the action planning system to choose and execute.

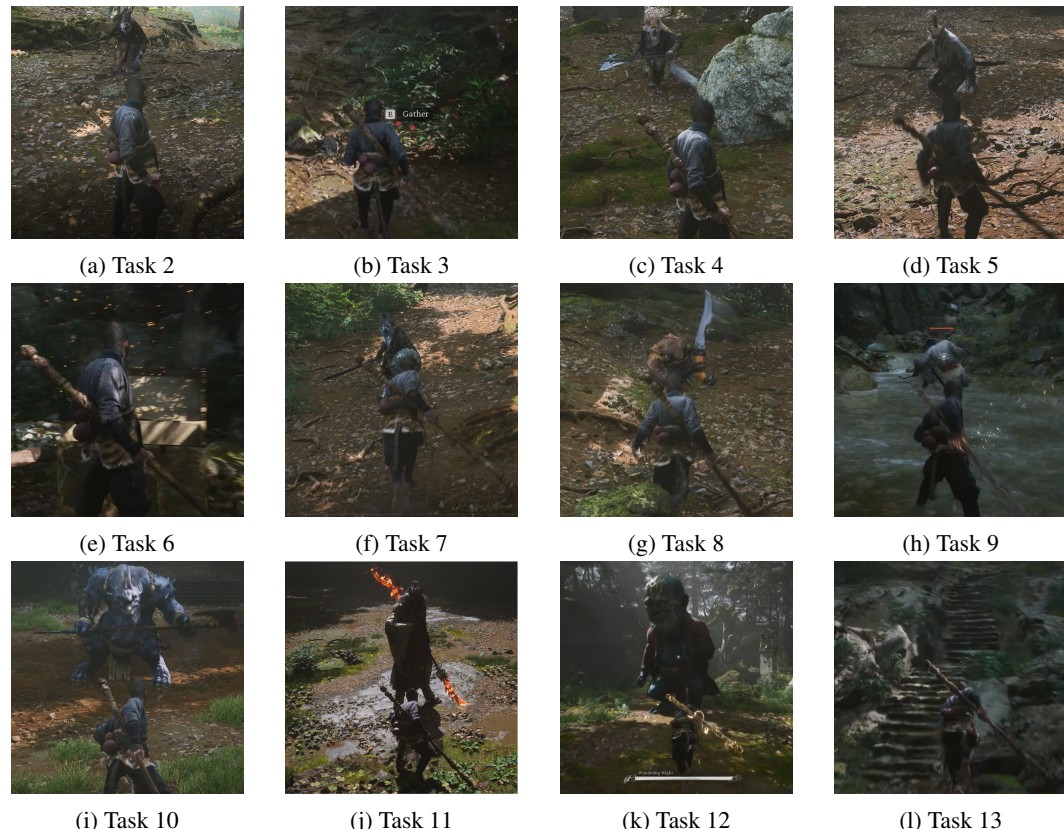

(a) Task 2      (b) Task 3      (c) Task 4      (d) Task 5

(e) Task 6      (f) Task 7      (g) Task 8      (h) Task 9

(i) Task 10      (j) Task 11      (k) Task 12      (l) Task 13

Figure 2: Image examples of defined tasks.

## 4 EXPERIMENTS

### 4.1 DATASET COLLECTION

We collected a human operation dataset that includes mouse and keyboard logs, as well as recordings of game screenshots. Specifically, we recruited 200 volunteers to play the BMW game and record their operations, with approximately 70% of them experiencing this game for the first time. To ensure the dataset's quality, we eliminated invalid data from volunteers who did not complete the tasks. Over the course of two weeks, we ultimately gathered a total of 1,000 valid data entries. Specifically, over 90% of tasks 11 and 12 (i.e., Defeat Guangzhi and Defeat Wandering Wight) were discarded, indicating that defeating these bosses in a single attempt poses a significant challenge for players. Moreover, we observed that volunteers exhibited redundant actions during the annotation process, such as excessive mouse clicks and scrolling. Therefore, some volunteers will be asked to play the game again to identify the optimal actions, and this refined data will be labeled as "clean" in our released dataset. Please refer to the supplementary material for more details about our dataset.

### 4.2 BENCHMARK AND TASK DEFINITION

To investigate the capabilities of existing VLMs in playing action games, we define 10 basic tasks and 3 challenging tasks aligned with the game's narrative, with 76.9% of these tasks occurring in combat scenes. As illustrated in Tab. 2 and Fig. 2, all tasks are concentrated in the first chapter of the game, due to the limited understanding and reasoning abilities of VLMs. In terms of bench-marking, we allow the agent to test each task 5 times and calculate the success rate for each task. For combat tasks, a task is deemed successful if the player's character defeats the enemy, while a task is considered a failure if the player's character is defeated and killed by the enemy. We have manually assessed the difficulty of 13 tasks, categorizing them as easy, medium, hard, and very hard. Due to the absence of maps and guidance, and the presence of numerous "invisible walls" in

Table 2: Task definitions in *Black Myth: Wukong* (BMW), where "*" indicates the challenging task.

| Task ID | Task Name | Description | Difficulty |
|---|---|---|---|
| 1 | Guidance | Defeat Erlang, the Sacred Divinty | Easy |
| 2 | Combat 1 | Defeat WolfScout | Easy |
| 3 | Gather | Gather | Easy |
| 4 | Combat 2 | Defeat WolfStalwart | Easy |
| 5 | Combat 3 | Defeat WolfSwornsword | Easy |
| 6 | Open | Open | Easy |
| 7 | Combat 4 | Defeat WolfSoldier | Easy |
| 8 | Combat 5 | Defeat Croaky | Easy |
| 9 | Combat 6 | Defeat Crow Diviner | Middle |
| 10 | Combat 7 | Defeat Bullguard | Hard |
| *11 | Combat 8 | Defeat Guangzhi | Very Hard |
| *12 | Combat 9 | Defeat Wandering Wight | Very Hard |
| *13 | Move | Autonomous Navigation | Very Hard |

the BMW game, we classify task 13, autonomous navigation (i.e., moving from the spawn point to the Bullguard's location within five minutes), as a very hard task. This is a challenging task even for human novices. We utilize the success rates from this benchmark to evaluate the performance of the VARP agent and various VLMs.

## 4.3 IMPLEMENTATION DETAILS

All evaluations are performed on a machine equipped with an NVIDIA RTX 4090 GPU running the Windows operating system. We use three of the most popular VLMs to drive our agent: GPT-4o-2024-05-13(OpenAI, 2024), Claude 3.5 Sonnet(Anthropic, 2024), and Gemini 1.5 pro(Google, 2024). We also utilize OpenAI's text-embedding-ada-002(Sonnet, 2022) model to generate embeddings for each action. The size of game interface for the BMW game is set to $1920 \times 1080$. During the inference of VLMs, we pause the game using the photo mode. We employ Grounding DINO(Liu et al., 2024) for object detection of people and objects in game screenshots to assist the VLMs in better extracting useful information.

## 4.4 PERFORMANCE EVALUATION

To evaluate the performance of the VARP agent without human guidance, we conducted experiments on our proposed benchmark while disabling the human-guided trajectory system of the VARP agent. In this performance evaluation, we only tested the benchmark and compared the VARP agent with human novice players.

We calculated the success rates of the VARP agent driven by different VLMs and human novice players when completing each task. As shown in Fig. 3, both the VARP agent and human novice players achieved high success rates on tasks 1 to 8, reaching nearly 100% on most tasks. In task 9, the VARP agent's average success rate was 40%, which also confirms its "middle" difficulty. The enemy in task 10 is the first boss-level monster that the player encounters in the game. For human novice players, the success rate for this task was 15.63%, while the VARP agent's average success rate was 20%. Task 11 and 12 are classified as "very hard," so the success rates for both human novices and the VARP agent were very low. Specifically, the VARP agent is limited by the reasoning speed of VLMs, making it unable to input every game frame in real-time and only able to input keyframes at second-level intervals. In ARPGs, this can easily result in missing critical information about enemy attacks. Therefore, task 11 and 12 are particularly challenging for the agent. In terms of autonomous navigation, humans can easily find the final boss enemy of the level within five minutes, but for VLMs, this is an almost impossible task. Without human guidance, the success rate is 0%. Since the game provides no guidance or hints for navigation tasks and contains many "invisible walls," VLMs lack the ability to perceive the correct path in the 3D scene without human assistance.

To explore the performance boundary of the VLM-based VARP agent, we chose a very-hard task and used the health of the enemy "Guangzhi" in task 11 as a metric. Due to the constraints of GPT-4o's maximum token count and inference time, we were unable to input all the video frames, which

led to the VARP agent being unable to defeat the highly aggressive "Guangzhi". However, during the combat, the VARP agent was capable of reducing the enemy's health by an average of about 40%, which may have reached the ability boundary of VLM for purely visual ARPG gameplay.

In summary, the VARP agent's performance on tasks 1 through 12 is already close to that of novice human players. However, in terms of 3D scene perception and prior knowledge, the VARP agent is still far inferior to humans.

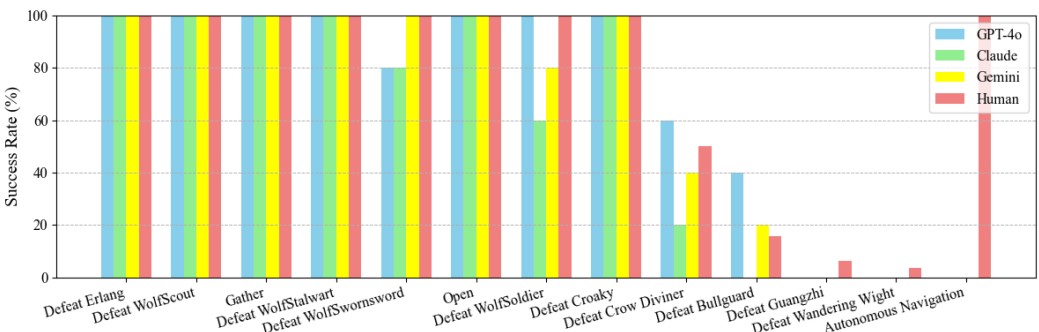

Figure 3: Evaluation results on various VLMs and human.

## 4.5 COMPARISION EVALUATION

To ensure a fair comparison, we adapted Cradle (Tan et al., 2024) to the BMW game. Specifically, we redesigned the predefined skills in Cradle to align with those of the VARP agent. We also revised the prompts for each module in Cradle to focus on combat tasks for the BMW game. Additionally, we used photo mode to pause the game. The VLM used in all experiments is GPT-4o. The experimental results are shown in Tab. 3.

In combat tasks, the success rate of VARP shows a significant improvement compared to Cradle, especially in medium and difficult tasks such as tasks 9-10. During the experiments, we found that Cradle tends to input a large amount of textual prompts into the VLM all at once during the decision-making process. This often leads to decision-making errors by the VLM, such as attempting to recover health when at full health, or forgetting to use spell skills when they are available. Furthermore, Cradle can only generate new actions based on textual prompts from game screenshots. However, in ARPG games, complex actions are a combination of various atomic operations, and there are no explicit textual prompts available for them. As a result, Cradle fails in medium and difficult tasks due to its inability to learn new actions.

Table 3: Comparison results with Cradle.

| Method | 1 | 2 | 3 | 4 | 5 | 6 | 7 | 8 | 9 | 10 |
|---|---|---|---|---|---|---|---|---|---|---|
| Cradle (Tan et al., 2024) | 100% | 80% | 100% | 60% | 60% | 100% | 40% | 100% | 20% | 0% |
| VARP | 100% | **100%** | 100% | **100%** | **80%** | 100% | **100%** | 100% | **60%** | **40%** |

## 4.6 ABLATION STUDY

To evaluate the effectiveness of the self-optimizable action generation module(SOAG) and the decomposable task-specific auxiliary module(DTSA) in the action planning system, we conducted experiments with each of these modules removed separately, calculating the success rate on the benchmark. The VLM used in this part of the experiment is GPT-4o-2024-05-13. As shown in Fig. 4, without SOAG, the agent's performance significantly declines in the middle and hard tasks. This is because the enemies in these tasks have high health points, resulting in prolonged battles. The function of SOAG is to continuously learn the enemies' attack patterns, aiding players in dodging and counterattacking. Therefore, in long-duration tasks like middle and hard tasks, the effectiveness of SOAG becomes more apparent. On the other hand, DTSA aims to decompose large tasks into smaller ones, focusing more on precision. This approach helps prevent global errors caused by local issues such as the forgetting and hallucination of the VLM. Hence, without DTSA, the agent tends to fail in some easy tasks.

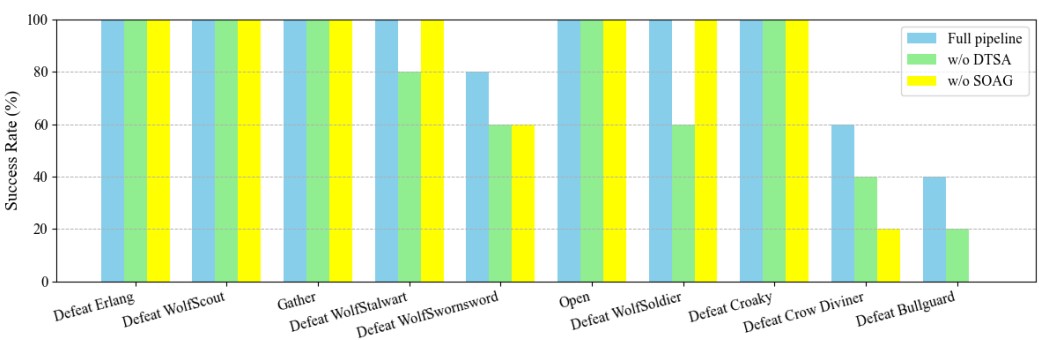

Figure 4: Ablation study.

## 4.7 CASE STUDIES

In this section, we will showcase some predefined actions and more newly generated actions with the corresponding game screenshots. For the VARP agent, these generated actions originate from two sources: one is the human-guided trajectory system, and the other is the SOAG in the action planning system. The VLM is GPT-4o.

To validate the effectiveness of the human-guided trajectory system, we introduced human guidance and conducted a case study on task 13, a task of very hard difficulty. The objective was for the VARP agent to control the player character to move from the "Earth Temple" spawn point to the location of the Bullguard enemy within 5 minutes. GPT-4o was chosen as the VLM. The experimental results showed a success rate of 40%. This indicates that human guidance can significantly enhance the decision-making accuracy of the agent. Fig. 5 shows the new action responsible for pathfinding generated by the human-guided trajectory system during the execution of task 13. Additionally, Fig. 5 depicts the new action generated by SOAG in response to the enemy, Bullguard, during the initial phase of this combat task. The enemy's current action indicates an impending attack: "swinging the axe downwards three times consecutively." Therefore, the new action should be to dodge consecutively more than three times before counterattacking.

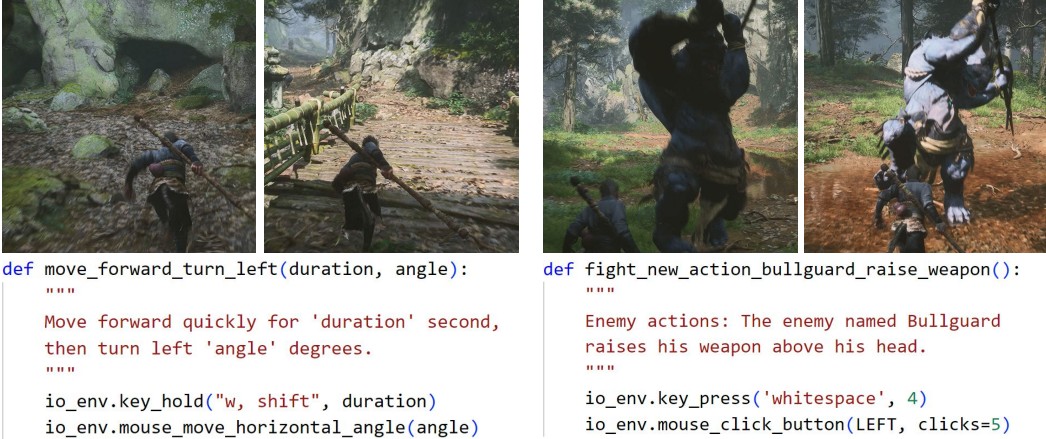

Figure 5: Simple cases of new actions generated by human guidance and SOAG.

As the combat task continues, the VARP agent can use or learn more complex actions to cope with more challenging scenarios. As shown in Fig.6, the actions in the first and second rows are predefined functions. The VARP agent automatically detects whether to use these actions based on the input visual information. For example, if the immobilization spell skill can be used, the agent executes the "fight_immobilization_spell_skill" action. Similarly, if the player's health is low, it uses the "recover_health" action.

The actions in the third row are generated by the human-guided trajectory system. Human prior knowledge can effectively guide the agent to improve efficiency in navigation tasks.

Figure 6: Complex cases of some actions and corresponding game screenshots.

The new actions in the fourth and fifth rows are summarized by SOAG after each combat interaction between the player character and the enemy and stored in the action library. These actions are specific to particular enemies and their attack patterns. For instance, in the fourth row, when the agent observes that an enemy named Bullguard is raising up his weapon, it indicates that the enemy is about to perform the action "chopping the axe downwards three times consecutively." The agent can then find a specific counter-action in the action library. At the beginning of the combat, this "fight_new_action_bullguard_raise_weapon" action is defined as dodging four times consecutively, followed by attacking five times, as shown in Fig.5. As the combat progresses, this action is optimized to counterattack during the intervals between dodges, significantly increasing the success rate and efficiency in defeating the enemy, as illustrated in the fourth row of Fig.6. This demonstrates that SOAG can continuously optimize the actions it generated.

## 5 CONCLUSION

In this study, we have explored the boundaries of current Vision Language Models (VLMs) in the context of complex action role-playing games (ARPGs) using "*Black Myth: Wukong*" as our experimental platform. Our proposed framework, VARP, introduces a novel approach to game interaction by leveraging visual-only inputs for action planning in ARPG environments. The VARP framework demonstrates its potential by achieving an 90% success rate in basic and moderate combat scenarios, suggesting that VLMs can be effectively utilized in tasks traditionally dominated by reinforcement learning. Our proposed benchmark can effectively evaluate the performance of visual-only agents in the BMW game. Additionally, the human operation dataset we provide offers a valuable resource for future research, enabling the study of human-like gameplay and action decision-making in visually complex environments. Our findings underscore the promise of multimodal agents in enhancing generalization and performance in action-oriented tasks within video games. Moving forward, the insights gained from this research could pave the way for more sophisticated agent designs that can handle a broader range of challenges in ARPGs and beyond.

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

# A    APPENDIX

## A.1    OVERVIEW

- Clarifications and Limitations (§A.2)
- Additional Related Work (§A.3)
- Additional Dataset Collection (§A.4)
- Additional Performance Evaluation (§A.5)
- Ethical Consideration (§A.6)
- Demo Video (§A.7)

## A.2    CLARIFICATIONS AND LIMITATIONS

Thank you for reading our paper. We would like to begin by clarifying and explaining some potential concerns that may arise due to the extreme popularity of the BMW game. We want to emphasize that our framework has broad applicability and will subsequently be generalized to include more games and other scenarios, not just limited to the BMW game. In this paper, we explored the potential of using VLMs to execute action combos in game tasks, particularly focusing on how it achieves victory against medium-powered monsters by leveraging the advantages of both action planning and visual trajectory modules. Additionally, we provided a human operation dataset, which presents possibilities for integrating technologies such as multi-modal retrieval-augmented generation, imitation learning, and reinforcement learning.

We must also candidly acknowledge some limitations in our research, specifically: 1) Task Definitions: As LLM- and VLM-based agents are still evolving, the current task definitions are somewhat simplistic. 2) Game Scenarios: Our research has only been tested within the BMW game and has not yet been extended to other scenarios. 3) Dataset Size: We have a limited amount of data, and in the future work, we plan to recruit more volunteers to collect higher quality data to enhance the depth of our research. 4) Model Capabilities: As shown in the performance evaluation section, there is still room for improvement in existing VLMs, including speed and accuracy. Therefore, it would be interesting to train an ARPG-specific VLM, such as VideoGameBunny[1].

Finally, we sincerely welcome your new ideas and feedback regarding this work, or even contribute your game records. Please feel free to reach out to us, and we look forward to exploring together, ultimately making VLMs play games as well as humans.

## A.3    ADDITIONAL RELATED WORK

### A.3.1    RL-BASED AGENTS IN ARPGS

Reinforcement learning (RL) has shown significant improvements in video games(Zhai et al., 2024; Bellemare et al., 2013; Kurach et al., 2020; Berner et al., 2019; Ellis et al., 2024; Samvelyan et al., 2019; Jaderberg et al., 2019; Qi et al., 2024; Wurman et al., 2022) especially action role-playing

---

[1]https://videogamebunny.github.io/

games (ARPG). DQN-play-sekiro (analoganddigital, 2021) employs the deep Q-network (DQN) algorithm to train AI to automate gameplay in "Sekiro: Shadows Die Twice." This project observes the game visuals and makes decisions based on the current state, gradually mastering the game strategy to defeat boss-level enemies. Additionally, AI achieves interactive learning based on reinforcement learning in "*Black Myth: Wukong*" by recognizing game images and scripting simulated keyboard input signals(Cat, 2024; fange, 2024). This method uses successful dodging as positive feedback while being attacked by monsters as negative feedback, prompting the AI to optimize its decision-making process.

These creative works not only showcase the potential of AI in complex gaming environments but also provide effective means for game testing and automated gameplay. However, agents trained solely using RL methods can only be applied to a limited range of specific tasks. For new tasks, the agent needs to be retrained. Therefore, agents based on this method have poor generalization capabilities.

## A.4  ADDITIONAL DATASET COLLECTION

We collected a total of approximately 1,000 valid data samples, each representing a video segment of a human completing a task along with the corresponding mouse and keyboard operation records. Among these, 4.0% represent task 1, 12.5% represent task 2, and so on. The specific information is shown in Fig. 7.

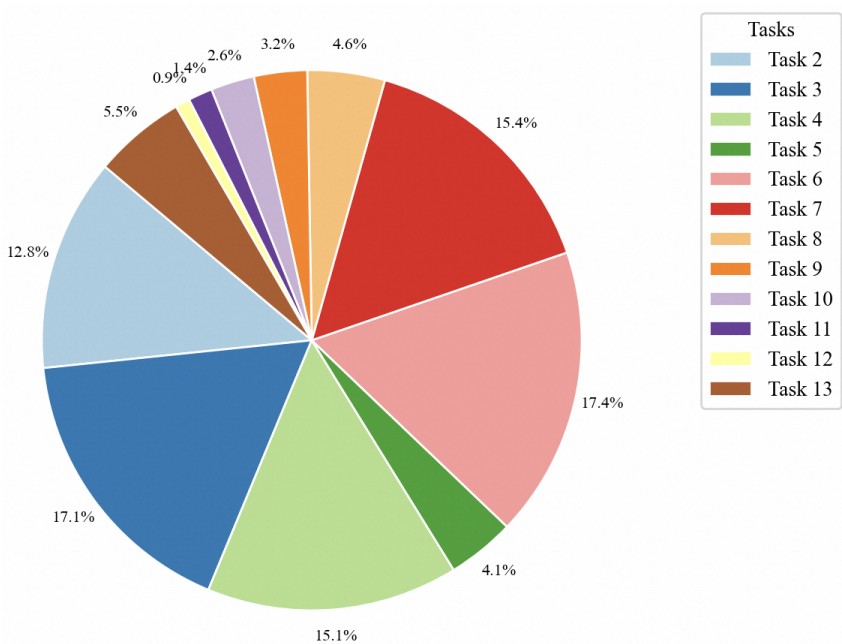

Figure 7: Dataset composition.

## A.5  ADDITIONAL PERFORMANCE EVALUATION

Based on Sec. 3.4 of the main text, we also recorded the average time and average inference count for the VARP agent without human guidance to complete each task. Each inference count represents the generation of an executable action, with each combat action containing an average of 8.6 atomic operations. Additionally, we recorded the number of atomic operations performed by human for each task. By dividing this number by 8.6, we estimated the inference count of human in combat tasks. As shown in Tab. 4, compared to humans, the VARP agent has a much lower inference count in task 1, task 9, and task 10. This indicates that humans tend to perform a large number of redundant operations when completing more difficult or time-consuming tasks in ARPGs, which is not conducive to task completion. In contrast, the actions generated by the VARP agent in these tasks are relatively more refined and concise.

Table 4: Additional evaluation results of the average time (in minutes) and average inference count.

| Task ID | GPT-4o (time) | Claude (time) | Gemini (time) | GPT-4o (count) | Claude (count) | Gemini (count) | Human(count) |
|---|---|---|---|---|---|---|---|
| 1 | 16.09 | 19.12 | 17.14 | 71.6 | 88 | 77 | **98.7** |
| 2 | 0.53 | 0.65 | 0.63 | 3.8 | 5 | 4.4 | 2.3 |
| 3 | 0.18 | 0.23 | 0.18 | 1.4 | 1.6 | 1.4 | 1.7 |
| 4 | 0.57 | 0.68 | 0.64 | 4.6 | 5.2 | 4.6 | 3.0 |
| 5 | 0.69 | 0.77 | 0.68 | 5.5 | 6.25 | 5.4 | 3.5 |
| 6 | 0.27 | 0.25 | 0.11 | 2 | 1.8 | 1.2 | 1.3 |
| 7 | 0.81 | 0.78 | 0.69 | 5.4 | 5.7 | 5.75 | 3.0 |
| 8 | 0.41 | 0.42 | 0.38 | 3.8 | 3.2 | 2.8 | 2.6 |
| 9 | 1.24 | 1.19 | 1.19 | 8.3 | 9 | 8.5 | **16.7** |
| 10 | 2.20 | - | 2.06 | 13.5 | - | 13 | **36.6** |

## A.6 ETHICAL CONSIDERATION

Our method can automatically play ARPGs, which may lead to game cheating and false advertising. This can have a significant negative impact on society. Therefore, it is crucial to consider methods that can reliably distinguish between genuine and forged content. We strongly condemn the unauthorized and malicious use of this technology and emphasize the need to consider ethical issues when using our method.

## A.7 DEMO VIDEO

We have provided a detailed demo video to demonstrate the effectiveness of our VARP agent. Please refer to https://varp-agent.github.io/.