# OpenReview forum: "Can VLMs Play Action Role-Playing Games? Take Black Myth Wukong as a Study Case"
_ICLR.cc/2025/Conference — Submitted to ICLR 2025_

### Official Review · Reviewer_bxWi · 2024-11-01

**Soundness:** 3
**Presentation:** 2
**Contribution:** 3
**Rating:** 5
**Confidence:** 4

**Summary:**

In this paper, the authors are exploring the use of visual language models (VLMs) to play action-role playing video games (ARPGs), specifically using one AAA title as a case study. They propose:
1) a VLM-based agent framework called VARP (Vision Action Role-Playing) which takes game screenshots as inputs and one of the 13 pre-defined tasks and outputs game actions (defined as combinations of atomic game actions)
2) a human-gameplay dataset with 1000 records, collected in-house with 200 mostly novice human players
3) an evaluation task set with 13 tasks specific to a AAA ARPG game Black Myth: Wukong (BMW), of different levels of difficulty: 9 easy, 1 medium, 1 hard and 3 very hard.

**Strengths:**

It is great to see the complexities of a AAA title being described and addressed, helping the research community make progress beyond simpler 2D environments. The authors do a good job of explaining their data collection process, mentioning the number of participants, as well as the compute resources utilised for the experiments. They also present results on ablating some of the key optimizations they added to the VARP framework to aid action selection.

The work is original in the sense that it addresses the use of VLMs for ARPGs, providing a framework which is able to update the action library, as opposed to using a static one, as highlighted in the comparison study against the closest baseline, Cradle. The authors are also open to share publicly the code and the datasets to contribute to the community and encourage reproducibility and extension upon their findings.

**Weaknesses:**

In evaluating VARP, authors highlight 2 limitations of using VLMs for action video games, such as their slow reasoning speed for the more difficult tasks, as well as the challenges they face when tasked with long horizon navigation queries. Given the premise in the title, I would have expected a more detailed critical analysis of the strengths and limitations of using VLMs used as agents in ARPGs.

Another opportunity to expand and strengthen the submission would be to present a clearer discussion on the value of the additional modules that distinguish it from the Cradle framework. Sections 4.5 and 4.6 are a great step in this direction, but I feel they could be further strengthened by including the very hard tasks. First by mentioning this in the context of related work, and secondly by considering the very hard tasks in the comparison with Cradle and the ablation study.

Even though the title does mention a case study on a specific game title, it would add a lot more strength to the submission to include a second game environment. One option would be for the authors to consider rephrasing the title. Having the initial question formulated as "Can VLMs play ARPGs?" and show insights from only one title, on a series of limited tasks, makes it harder to claim that the question is being comprehensively addressed.

It would be good to see a more detailed discussion on how the choice of VLM driving VARP makes a difference on the overall success rates. Along similar lines, in order to best support the community, it would be good to see stronger reasoning supporting the authors’ initial choice of VLM models - why were GPT-4o, Claude and Gemini selected in experiments, and not others? What makes them suitable for this type of tasks?

**Questions:**

Clarifying Questions:

-	In Section 3.2.1, could you elaborate more on the size of the predefined action set and link to some examples (such as the ones in Figures 5 and 6)? Also, it would be great to know more about the process of selecting those actions.
-	In Section 3.2.3, would the 5 submodules generalize to a wide range of ARPGs?
-	In Section 3.3, it would be good to clarify that the human-guided trajectory system is only being used for the very hard tasks.
-	In Section 4.4, could you specify what were the GPT-4o constraints in terms of maximum token count?
-	On the Defeat WolfSwornsword task, Gemini performed better than the other 2 VLMs, GPT-4o and Claude. Any intuition on why that was the case?
-	In Table 4 in the appendix, did you collect inference performance numbers on the very hard tasks as well that made use of the human-guided trajectory system? It would be interesting to understand the cost of running that additional component. Also, could you elaborate more on why is task 1 (easy) much slower compared to the others?
-	Was there an option considered to try out VARP on the Cradle dataset and tasks? It would be good to see if it exceeds the performance of Cradle.
-	In the abstract and conclusion, you mention a 90% success rate in basic and moderate combat scenarios, is this number based on success rates in Table 3? If so, the average for VARP should be 88%.

Minor comments/Suggestions:

-	In the introduction of Section 3.2, it would be good to point to which respective section will update the situation library (Section 3.2.1), which one will define the updatable action library (Section 3.2.2) and which one will introduce the self-optimizable action generation module (Section 3.2.3).
-	In section 3.2.1, for readability, you can mention that you are about to start introducing the 5 basic modules from Cradle.
-	In Section 3.3, it would be good to maybe add in the appendix examples of game screenshots and human operation pairs collected in the human dataset.
-	In Figure 2, it would be best to also add the short description of each task as the description for each subfigure/example.
-	In Figure 3, for readability, it would be good to add the numerical task index along with the description (as mapped and defined in Table 2), to make it easier for the reader to keep track of the task numbers referred to in the main body of the paper.
-	In Section 4.4, I would add a link to the additional inference evaluation added in the appendix A.5.
-	It would be good to add a more descriptive caption to Table 3, specifying that the metric depicted in the comparison is the success rate.
-	In Section  4, for consistency, consider standardizing the use of the term GPT-4o to reflect the use of this specific VLM. In some places, it is referred to as GPT-4o and in some, it is referred to as GPT-4o-2024-05-13.
-	For readability, in Figure 5, it would be best to split it into 2 subfigures, (a) for the pathfinding action generated by the human-guided trajectory system and (b) for the action generated by the SOAG system.
-	Similarly, Figure 6 could be more descriptive, highlighting which types of actions are predefined and which are generated.

Minor Corrections:

-	in the introduction (lines 90 and 104), there are 12 tasks mentioned, with 75% of them focused on combat. In the abstract and the Experiments section, there are 13 tasks described, with a classification of 76.9% combat.
-	Noted 3 typos: one on line 100, word “easy” is misspelled, one on line 176, the word “we” should be written starting with lowercase, one on line 259 in “in Sec 4.1” instead of 3.1.
-	In Section 4.7, line 454, the choice of GPT-4o model is repeated. It was stated in the same section on line 450.

---

### Official Review · Reviewer_8C1w · 2024-11-02

**Soundness:** 3
**Presentation:** 3
**Contribution:** 2
**Rating:** 5
**Confidence:** 3

**Summary:**

This paper introduces a VLM-based agent that can play a AAA action role-playing (ARPG) game “Black Myth: Wukong” (BMW). Notably, The system features a human-guided trajectory system that generates an action library based on human gameplay data. The contribution includes defining 13 game-playing tasks in BMW, releasing a BMW gameplay dataset, and introducing a new framework for playing ARPG games based on VLM. The experiment shows the introduced VLM-based agent outperforms Cradle, a general-purpose computer control agent, and is competitive against human players.

**Strengths:**

This paper not only introduces an agent framework but also introduces a benchmark and a dataset. Notably, the dataset is collected by recruiting 200 human players to play the game. The proposed agent generates an action library from human gameplay data, does not rely on text-based guiding information. The agent directly takes screenshots as input and output mouse and keyboard commands, and achieves good performance in the 13 introduced tasks.

**Weaknesses:**

The significance of this work is not so clear. According to the presented results, Cradle, a general computer control agent based on VLMs, can also play BMW. Though the results show the proposed VARP outperforms Cradle, to my understanding, the Cradle framework does not use human gameplay data and is designed for general purposes. It is not surprising that the proposed agent can outperform Cradle.
Meanwhile, there are some concerns about the experiment part:
1. The proposed agent is not compared to the RL-based agent for BMW, i.e., “Other project” in Tab. 1 (I guess it’s AI-Wukong), which is also an agent playing BMW
2. How many trials do the authors repeat for each task? It seems that all success rates are divisible by 10%, so perhaps all tasks are tested for 10 trials. If possible, I would recommend the authors test agents for more trials, or explain why 10 trials are enough/why not test more trials.

Minor issues:

Lots of opening brackets are not separated from the texts with a space. For example, “action role-playing games(ARPG),”; “VARP(Vision Action Role-Playing)”.

**Questions:**

1.	How does AI-Wukong’s performance compare to VARP? Or is there some reason for not comparing AI-Wukong with VARP?
2.	What is the practical impact of VARP, for example, how can VARP potentially benefit the game industry?

---

### Official Review · Reviewer_oSF8 · 2024-11-04

**Soundness:** 2
**Presentation:** 4
**Contribution:** 2
**Rating:** 5
**Confidence:** 3

**Summary:**

The paper introduces a new eval for Multimodal LLMs (those which accept images). The authors take the Action RPG “Black Myth: Wukong” and produce a benchmark of 13 tasks from the game. The game produces screenshots. The environment asks agents to produce high level actions which are transformed by python functions into specific actions of the model.

The authors collect human performance on the task and produce a dataset of expert trajectories. This is used as a retrieval system during the agents playthrough

The authors also simultaneously propose a method for composing LLMs to approach these problems (VARP agent). The VARG agent, using Gemini / Claude / GPT-4-Turbo is able to solve 7/13 tasks completely.

**Strengths:**

The paper is well presented and the task is original

The writing is clear

The 5 final tasks are clearly difficult

Good ablations to identify what works well

**Weaknesses:**

Given the baseline (Cradle) is also able to perform well on the 5 easy tasks this reduces the usefulness of those tasks.



My biggest concern is that i’m not sure how performance on this evaluation relates to real world capabilities. For example montezuma’s revenge was directly related to hard exploration or hinabi related to identify cooperative policies.  I think some framing to explain what models failing on the final 5 tasks demonstrates.

I think a strong evaluation protocol could be suggested - e.g. have held out tasks or tasks that composed of easier tasks. This would help interpret what models fail at. In particular currently this seems to be a task that measures in-distribution performance, with accessing to expert trajectories.

The dataset collected is largely skewed towards similar task - given the first 5 are not as useful - how big is the actual dataset?


The Decomposable Task-Specific Auxiliary (DTSA) is very tailored to the game (as mentioned in the limitations).

I think most of the performance is coming from the human-guided trajectory system. Why was removing this subcomponent not part of the ablation?

**Questions:**

You mention both in abstract and introduction that 75% of the tasks are combat based - could you explain why this is important?

Some typos - you say you define 13 tasks in the abstract but 12 in the introduction.

Figure 7 does not have task 1

---

### Official Review · Reviewer_6yr1 · 2024-11-07

**Soundness:** 2
**Presentation:** 2
**Contribution:** 2
**Rating:** 5
**Confidence:** 4

**Summary:**

This work builds an AI agent framework, aiming to deal with Action Role-Playing Games. It successfully demonstrates the huge potential of LMM in decision-making. In addition, it builds a benchmark that may be useful for the future research in this filed.

**Strengths:**

1. This work needs lots of engineering efforts and the proposed benchmark will be handy for future research.
2. This work explores some of the potential of existing large models, offering the audience plenty of room for imagination.

**Weaknesses:**

1. The novelty of this work seems very low. The framework is similar to Cradle and many other AI agent works, as well as some tailored modules, mainly in section 3.2.3, for BMW.
2. The game is not an open-ended world and the skill library can be enumerated. The level of difficulty is still somewhat limited.

**Questions:**

I greatly appreciate these AI agent works as they explore the boundaries of VLM capabilities. However, from a methodological point, their contributions to academia are quite limited. Therefore, I believe this type of work should be reorganized into a benchmark-focused effort, dedicated to advancing the field.

---

### Meta-Review · Area_Chair_T1cK · 2024-12-17

**Metareview:**

This paper proposes to benchmark VLM agents on games, a promising direction to test agentic reasoning capabilities in a sandbox environment. The reviewers raised several solvable concerns, but the authors did not engage. Thus, the paper remains below the threshold, with encouragement to the authors to make these improvements for the next deadline.

**Additional Comments On Reviewer Discussion:**

No discussion!

---

### Decision · Program_Chairs · 2025-01-22

Reject